# Structural Analysis of the ESCRT-III Regulator Lethal(2) Giant Discs/Coiled-Coil and C2 Domain-Containing Protein 1 (Lgd/CC2D1)

**DOI:** 10.3390/cells13141174

**Published:** 2024-07-10

**Authors:** Thomas Breuer, Christine Tibbe, Tobias Troost, Thomas Klein

**Affiliations:** Institute of Genetics, Heinrich-Heine-Universitaet Duesseldorf, Universitaetsstr. 1, 40225 Duesseldorf, Germany; thomas.breuer@hhu.de (T.B.); christine.tibbe@web.de (C.T.); tobias.troost@uni-duesseldorf.de (T.T.)

**Keywords:** Lgd/CC2D1A/CC2D1B, ESCRT machinery, Shrub/CHMP4/Snf7, Notch signalling

## Abstract

Members of the LGD/CC2D1 protein family contain repeats of the family-defining DM14 domains. Via this domain, they interact with members of the CHMP family, which are essential for the ESCRT machinery-mediated formation of intraluminal vesicles during endosome maturation. Here, we investigate the requirement of the DM14 domains for the function of Lgd in detail. We found that although both odd-numbered DM14s can act in a functionally redundant manner, the redundancy is not complete and both contribute to the full function of Lgd. Our analysis indicates that some of the AAs that form the KARRxxR motif of the onDM14s are not exchangeable by similarly charged AAs without loss of function, indicating that they not only provide charge, but also fulfil structural roles. Furthermore, we show that the region of Lgd between DM14-4 and the C2 domain as well as its C-terminal region to the C2 domain are important for protein stability/function. Moreover, we analysed the importance of AAs that are conserved in all DM14 domains. Finally, our analysis of the *C. elegans* ortholog of Lgd revealed that it has only one DM14 domain that is functionally equivalent to the onDM14s. Altogether, the results further the understanding of how Lgd family members regulate the ESCRT machinery.

## 1. Introduction

Membrane remodelling is a crucial cellular process that underlies several processes, such as the formation of the intraluminal vesicles of the maturing endosome by the ESCRT machinery. The ESCRT machinery is involved in many cellular processes where membrane regions are abscised away from the cytosol [1]. Initially, the machinery was investigated during the formation of intraluminal vesicles (ILVs) at the limiting membrane of the maturing endosome (ME). During ILV formation, the machinery comprises five complexes acting in sequence, ESCRT-0, -I, -II, -III, and an oligomeric complex of the AAA-ATPase Vps4 [1]. The early-acting complexes concentrate cargo, which is recognised through its ubiquitin label at sites of ILV formation. ESCRT-III is the centrepiece that, together with Vps4, mediates the membrane abscission process. ESCRT-III consists of four members of the CHMP family, which assemble into a filament at the limiting membrane of the ME. Vps20 initiates the polymerisation of CHMP4B, encoded by Shrub in *Drosophila* and Snf7 in yeast, into a homopolymer that is modified in some way by Vps24, Vps2, and later-acting ESCRT-III members such as CHMP5, Did2 and Ist-1.

Members of the CHMP family fold into helical hairpins and cycle between the cytosolic monomeric and polymeric membrane-bound state [1]. It has been shown that Shrub/CHMP4B possesses two opposing electrostatic surfaces that mediate the electrostatic interactions between protomers of the filament at membranes [2,3]. How the assembled ESCRT-III filament performs the reverse topology membrane abscission is not understood. Recent in vitro experiments suggest that the assembled Shrub filament either serves as a template for the formation of more rigid filaments made of other CHMP members, or pinches off membranes by itself with help of the other CHMP members [4]. The Shrub filaments appear to be constantly remodelled by the action of Vps4, which can remove and replace protomers from the filament under ATP consumption [4,5].

Work so far has revealed that the *Drosophila* tumour suppressor lethal (2) giant discs (Lgds) and its two mammalian orthologs coiled-coil and C2 domain-containing protein 1a (Cc2d1a/Lgd2) and 1b (Cc2d1b/Lgd1) are positive regulators of the activity of the ESCRT machinery [6,7,8,9]. A consequence of loss of function of *lgd* in *Drosophila* is the ligand-independent activation of the Notch pathway, as well as the tissue-specific prolonged activation of the Dpp/BMP pathway [10,11]. In addition, it results in the defective cytokinesis of germline cells [12]. In mammals, the loss of CC2D1a results in death shortly after birth in mice and intellectual disability and autism in humans [13]. The loss of CC2D1b/LGD1 causes a defect in the formation of the nuclear envelope after cleavage of cell culture cells [14]. However, the complete knock-out of Lgd1 does not affect development or vitality in mice, suggesting that either the loss is compensated by alternative pathways or Lgd2 can act in a functionally redundant manner [15]. In favour of the second possibility, the concomitant loss of function of both CC2D1/LGD genes results in early embryonic lethality, indicating that they function in a redundant manner [16]. Cell culture experiments addressing molecular function of the Lgd orthologs in humans reveal a variety of functions ranging from regulation of cell signalling to cytokinesis [13]. 

Lgd proteins possess four tandem repeats of the family-defining *Drosophila melanogaster* 14 (DM14) domain, followed by a phospho-lipid-binding C2 domain ([17,18,19], Figure 1A,A’). The DM14 domains are helical hairpins that mediate protein–protein interactions with members of the CHMP family, such as Shrub/CHMP4, CHMP2 and CHMP7 ([8,9,14,20], Figure 1A,D,E,F). Lgd physically interacts with Shrub via its two odd-numbered DM14 domains (onDM14s) [21]. In contrast to the two even-numbered domains, the onDM14s possess an extended positively charged surface and appear to function in a redundant manner, shown by the finding that Lgd variants with only one of the odd-numbered domains can completely rescue *lgd*-null mutants if already present in one copy [21] (Figure 1D–F’). However, they fail to rescue a sensitised background where a copy of functional *shrub* is removed in addition to *lgd* (sensitised background, genotype: *lgd*/*lgd*; *shrub/*+, see Figure 1B–C’). This indicates that the functional redundancy among the onDM14s is required to tolerate the *lgd shrub* double heterozygous situation. Whether the redundancy of the odd-numbered domains is complete or whether they also have individual functions is not known. The findings for Lgd and Shrub have been recently confirmed for the orthologs of mammals and *C. elegans* [2,7]. 

The interaction of the third DM14 domain (DM14-3) with Shrub has been determined at the atomic level, revealing that the positively charged surface of DM14-3 engages in electrostatic interactions with the negatively charged surface of Shrub [2,3]. It also identified the amino acids (AAs) responsible for binding, which are conserved in both onDM14s of Lgd. Replacement of individual AAs of this motif, here defined as the KARRxxR motif (see Figure 2), by Alanine (A) abolishes the function of the onDM14s. The meaning of the KARRxxR motif has also been confirmed for the mammalian LGD orthologs [2]. Although essential, it is not clear whether the motif is sufficient for the functionality of the onDM14s. Another open question is whether the positively charged AAs of the KARRxxR motif only provide charge, or an additional function. These questions are important to answer for the understanding of the function of DM14 domains and their interaction with members of the CHMP family.

Since the binding between Shrub protomers and the binding of Shrub to an onDM14 both require the negatively charged surface of Shrub, they are mutually exclusive. Nevertheless, recent work demonstrated that Lgd is a positive regulator of Shrub function, since its loss of function strongly reduces the activity of Shrub [6,7,9]. One likely possibility for how Lgd acts is that its complex formation with Shrub prevents inappropriate polymerisation of Shrub in the cytosol. The inappropriate polymerisation would reduce the function of Shrub at the endosomal membrane [6]. Another recently suggested possibility is that Lgd is required to efficiently recruit Shrub to the LM of the ME [7].

Here, we investigate the requirement of the DM14 domains for the function of Lgd in detail. We found that although both onDM14s can act in a functionally redundant manner, the redundancy is not complete and both contribute to the full function of Lgd. Our analysis indicates that some of the AAs that form the KARRxxR motif of the onDM14s are not exchangeable by similarly charged AAs without loss of function, indicating that they not only provide charge but also fulfil structural roles. Moreover, we analysed the importance of AAs that are conserved in all DM14 domains. Furthermore, we show that the region of Lgd between DM14-4 and the C2 domain as well as its C-terminal region to the C2 domain are important for protein stability/function. Finally, our analysis of the *C. elegans* ortholog of Lgd revealed that it has only one DM14 domain that is functionally equivalent to the onDM14s. Altogether, the results further the understanding of how Lgd family members regulate the ESCRT machinery.

## 2. Materials and Methods

### 2.1. Fly Stocks

Gbe+Su(H)-lacZ [23], *wg*-lacZ [24], *lgd^d7^* FRT40A [25], and *shrub^4^^-^^1^* FRTG13 [26] were used in this study. Flies were raised on a standard diet and kept at room temperature. For crossings, flies were raised at 25 °C. 

### 2.2. Generation of Constructs and Vectors

Lgd rescue constructs were expressed under the control of the endogenous Lgd promotor (*lgdP*) as described before [9]. The used vector harbours the proximal lgd genomic elements (548 bp upstream and 553 bp downstream of the lgd ORF).

Truncated Lgds were generated using Gibson Assembly with *pattB-lgdP-Lgd-HA* as a template [9]. For the Gibson Assembly, primers were used to generate gene fragments with overlaps of approximately 30 bp. Primer sequences are in the Appendix A as Appendix A. AA changes were achieved by site-directed mutagenesis. 

All *lgdP* constructs were inserted into the genomic attP landing site 86Fb. Injection of embryos was either performed in-house or by BestGene Inc. (Chino Hills, CA 91709, USA).

### 2.3. Genotypes 

**Figure 1B:** *w*; *+/+*; *Gbe+Su(H)-lacZ/+*; **Figure 1C:** *w*; *lgd^d7^ FRT40A/lgd^d7^ FRT40A*; *Gbe+Su(H)-lacZ/+*; **Figure 2B:** *w*; *lgd^d7^ FRT40A/lgd^d7^ FRT40A*; *Gbe+Su(H)-lacZ/lgdP.lgd-1-2-3-4-C2 86Fb*; **Figure 2C:** *w*; *lgd^d7^ FRT40A/lgd^d7^, shrb^4-1^ FRTG13*; *Gbe+Su(H)-lacZ/lgdP.lgd-1-2-3-4-C2 86Fb*; **Figure 2D:** *w*; *lgd^d7^ FRT40A/lgd^d7^ FRT40A*; *Gbe+Su(H)-lacZ/lgdP.lgd-3-4-C2 86Fb*; **Figure 2E:** *w*; *lgd^d7^ FRT40A/lgd^d7^, shrb^4-1^ FRTG13*; *Gbe+Su(H)-lacZ/lgdP.lgd-3-4-C2 86Fb*; **Figure 2F:** *w*; *lgd^d7^ FRT40A/lgd^d7^ FRT40A*; *Gbe+Su(H)-lacZ/lgdP.lgd-1-C2 86Fb*; **Figure 2G:** *w*; *lgd^d7^ FRT40A/lgd^d7^, shrb^4-1^ FRTG13*; *Gbe+Su(H)-lacZ/lgdP.lgd-1-C2 86Fb*; **Figure 2H:** *w*; *lgd^d7^ FRT40A/lgd^d7^ FRT40A*; *Gbe+Su(H)-lacZ/lgdP.lgd-1-1-C2 86Fb*; **Figure 2I:** *w*; *lgd^d7^ FRT40A/lgd^d7^, shrb^4-1^ FRTG13*; *Gbe+Su(H)-lacZ/lgdP.lgd-1-1-C2 86Fb*; **Figure 3C:** *w*; *lgd^d7^ FRT40A/lgd^d7^ FRT40A*; *Gbe+Su(H)-lacZ/lgdP.lgd-3 K387R -4-C2 86Fb*; **Figure 3D:** *w*; *lgd^d7^ FRT40A/lgd^d7^, shrb^4-1^ FRTG13*; *Gbe+Su(H)-lacZ/lgdP.lgd-3 K387R -4-C2 86Fb*; **Figure 3E:** *w*; *lgd^d7^ FRT40A/lgd^d7^ FRT40A*; *Gbe+Su(H)-lacZ/lgdP.lgd-3 R389K-4-C2 86Fb*; **Figure 3F:** *w*; *lgd^d7^ FRT40A/lgd^d7^ FRT40A*; *Gbe+Su(H)-lacZ/lgdP.lgd-3 R390K-4-C2 86Fb*; **Figure 3G:** *w*; *lgd^d7^ FRT40A/lgd^d7^ FRT40A*; *Gbe+Su(H)-lacZ/lgdP.lgd-3 R393K-4-C2 86Fb*; **Figure 3H:** *w*; *lgd^d7^ FRT40A/lgd^d7^ FRT40A*; *Gbe+Su(H)-lacZ/lgdP.lgd-3 A388G-4-C2 86Fb*; **Figure 3I:** *w*; *lgd^d7^ FRT40A/lgd^d7^, shrb^4-1^ FRTG13*; *Gbe+Su(H)-lacZ/lgdP.lgd-3 A388G-4-C2 86Fb*; **Figure 4F:** *w*; *lgd^d7^ FRT40A/lgd^d7^ FRT40A*; *Gbe+Su(H)-lacZ/lgdP.lgd-4-C2 86Fb*; **Figure 4G:** *w*; *lgd^d7^ FRT40A/lgd^d7^ FRT40A*; *Gbe+Su(H)-lacZ/lgdP.lgd-4 KARRxxR HA 86Fb*; **Figure 4H:** *w*; *lgd^d7^ FRT40A/lgd^d7^ FRT40A*; *Gbe+Su(H)-lacZ/lgdP.lgd-2-2-HA 86Fb*; **Figure 4I:** *w*; *lgd^d7^ FRT40A/lgd^d7^ FRT40A*; *Gbe+Su(H)-lacZ/lgdP.lgd-2KARRxxR-4 KARRxxR -C2 86Fb*; **Figure 4J:** *w*; *lgd^d7^ FRT40A/lgd^d7^ FRT40A*; *Gbe+Su(H)-lacZ/lgdP.lgd 1-2 KARRxxR-3-4 KARRxxR -C2 86Fb*; **Figure 5B:** *w*; *lgd^d7^ FRT40A/lgd^d7^ FRT40A*; *Gbe+Su(H)-lacZ/lgdP.lgd-3-R368A-C2 86Fb*; **Figure 5C:** *w*; *lgd^d7^ FRT40A/lgd^d7^ FRT40A*; *Gbe+Su(H)-lacZ/lgdP.lgd-3-K380A-4-C2 86Fb*; **Figure 5D:** *w*; *lgd^d7^ FRT40A/lgd^d7^ FRT40A*; *Gbe+Su(H)-lacZ/lgdP.lgd-3 K396A-4-C2 86Fb*; **Figure 5E:** *w*; *lgd^d7^ FRT40A/lgd^d7^ FRT40A*; *Gbe+Su(H)-lacZ/lgdP.lgd 3G408A-4-C2 86Fb*; **Figure 5F:** *w*; *lgd^d7^ FRT40A/lgd^d7^ FRT40A*; *Gbe+Su(H)-lacZ/lgdP.lgd-3 P417A-4-C2 86Fb*; **Figure 5G:** *w*; *lgd^d7^ FRT40A/lgd^d7^, shrb^4-1^ FRTG13*; *Gbe+Su(H)-lacZ/lgdP.lgd-3 R368A-4-C2 86Fb*; **Figure 5H:** *w*; *lgd^d7^ FRT40A/lgd^d7^, shrb^4-1^ FRTG13*; *Gbe+Su(H)-lacZ/lgdP.lgd-3 K380A-4-C2 86Fb*; **Figure 5I:** *w*; *lgd^d7^ FRT40A/lgd^d7^, shrb^4-1^ FRTG13*; *Gbe+Su(H)-lacZ/lgdP.lgd-3-4-C2 86Fb*; **Figure 6E:** *w*; *lgd^d7^ FRT40A/lgd^d7^ FRT40A*; *Gbe+Su(H)-lacZ/lgdP.lgd-Δ558-664 HA 86Fb*; **Figure 6F:** *w*; *lgd^d7^ FRT40A/lgd^d7^ FRT40A*; *Gbe+Su(H)-lacZ/lgdP.lgd-LAP HA 86Fb*; **Figure 6G:** *w*; *lgd^d7^ FRT40A/lgd^d7^ FRT40A*; *Gbe+Su(H)-lacZ/lgdP.lgd-ΔC HA 86Fb*; **Figure 7B:** *w*; *lgd^d7^ FRT40A/lgd^d7^ FRT40A*; *Gbe+Su(H)-lacZ/lgdP.C.e.Lgd 86Fb*; **Figure 7C:** *w*; *lgd^d7^ FRT40A/lgd^d7^, shrb^4-1^ FRTG13*; *Gbe+Su(H)-lacZ/lgdP. C.e.Lgd 86Fb*; **Figure 7D:** *w*; *lgd^d7^ FRT40A/lgd^d7^ FRT40A*; *Gbe+Su(H)-lacZ/lgdP. Δ 1-135 HA 86Fb*; **Figure 7E:** *w*; *lgd^d7^ FRT40A/lgd^d7^, shrb^4-1^ FRTG13*; *Gbe+Su(H)-lacZ/lgdP.lgd-Δ 1-135 HA 86Fb.*

### 2.4. Immunohistochemistry and Microscopy

Dissected wing imaginal discs of wandering L3 larvae were fixed with 4% paraformaldehyde in PBS (4%PFA) for 30 min and washed with PBS. Permeabilization and blocking were carried out using 0.3% Triton X-100 in PBS (PBT) and 5% normal goat serum (NGS) for 30 min at room temperature. Primary antibody incubation was performed in 5% NGS in 0.3% PBT for 2 h at room temperature followed by three washing steps with PBT. The corresponding secondary antibody was applied in 5% NGS in 0.3% PBT for 2 h at room temperature. The following antibodies were used: mouse anti-Wg 4D4 (1:10, Developmental Studies Hybridoma Bank (DSHB), Iowa City, IA, USA), rabbit anti-β-Gal (polyclonal) (1:5000, MP Biomedicals, LLC., Solon, OH, USA). Fluorophore-conjugated secondary antibodies were purchased from Invitrogen. Nuclei staining was carried out using Hoechst 33258 dye. 

Images were acquired with the Zeiss Axio Imager Z1 Microscope equipped with a Zeiss Apotome or Apotome2 (Carl Zeiss Microscopy GmbH, Jena, Germany).

### 2.5. Western Blot Analysis

For comparison of protein levels of different expressed Lgd variants, lysates of wandering L3 larvae were analysed by Western blot. For that, the larvae were collected and washed in PBS and then transferred in lysis buffer (10% Glycerin; 50 mM HEPES (pH7.5); 150 mM NaCl; 0.5% Triton-X-100; 1.5mM MgCl_2_; 1mM EGTA; Protease Inhibitor Cocktail (Sigma-Aldrich, St Louis, MI, USA). Rupture of the larval tissues was then carried out by using a micro pestle. After 15 min of incubation on ice with lysis buffer, the lysates were centrifuged and the supernatant was collected in a fresh tube avoiding the transfer of excess fat. Laemmli Buffer was added and heated to 95 °C for 10 min. 

Lysates were loaded on a 10% SDS-PAGE gel and blotted on a PVDF membrane. Immunostaining was performed using standard protocols. Membranes were blocked using 5% milk powder in PBS and immunostaining was carried out in 2% milk powder in PBS.

The following antibodies were used: rat anti-HA (1:3000; 3F10, Roche) and rabbit anti-actin (1:10,000, 17H19L35, Invitrogen, Waltham, MA, USA). HRS-conjugated secondary antibodies were purchased from Jackson Immuno Research (872 W Baltimore Pike, West Grove, PA, USA). Chemiluminescence was detected using WesternBright ECL HRP substrate (Advansta, 2140 Bering Dr, San Jose, CA, USA) and the Amersham ImageQuant 800 (Cytiva, Marlborough, MA, USA). 

### 2.6. Sequence Alignments and Structure Analysis and Visualisation

Sequence alignments were carried out using CLC Genomics Workbench 20 (QIAGEN, Aarhus, Denmark). The predicted Lgd protein structure was used from the Alphafold database (ID: AFQ9VKJ9-F1). Structure analysis and visualization were achieved using PyMOL (Schrödinger Inc, New York, NY, USA). Protein Structure Source: Figure 1A: Alphafold Database, Figure 1E,F: Alphafold prediction (Colabfold), Figure 1F’,F’ Alphafold multimere prediction (Colabfold); Figure 4A–C’: Alphafold prediction (Colabfold); Figure 5J: Alphafold Database; Figure 6A,B: Alphafold Database, Figure 6C: Alphafold prediction (Colabfold).

## 3. Results

### 3.1. Partial Functional Redundancy of the onDM14 Domains of Lgd

Loss of *lgd* function causes the ligand-independent activation of the Notch pathway in all cells of the imaginal discs [10,17,18,19]. This is revealed by the expanded expression of the target gene *wingless* (*wg*) and the Notch activity reporter Gbe+Su(H) in the wing primordium (Figure 1B–C’). We used the expression of the mentioned target genes as a read out of the functionality of the Lgd variants investigated herein through rescue assays. All variants are expressed under the control of the endogenous *lgd* promoter (*lgd*P) and inserted into the same genomic landing site to guarantee similar expression levels [9]. We previously showed that a variant with only one or two copies of DM14-3 (Lgd-3-C2 and Lgd-3-3-C2, instead of Lgd-1-2-3-4-C2) can completely rescue the *lgd* mutant, but not the sensitised background (*lgd*+/*lgd*
*shrub*) [9,21]. In the sensitised background, weak ectopic expression is still observed, indicating that the rescue is only partial [9,21]. However, the partial rescue by Lgd-3-3-C2 was better than by Lgd-3-C2 [21]. In contrast, a variant consisting of both odd-numbered DM14 domains, Lgd-1-3-C2, completely rescued the sensitised background. This indicates that DM14-1 is also important for the full function of Lgd [21]. 

To further explore the function of DM14-1, we tested the rescue abilities of Lgd-1-C2 and found that, similar to Lgd-3-C2 and Lgd-3-3-C2, it can fully rescue *lgd* mutants, but not the sensitised background (Figure 2B–G’). Nevertheless, the rescue of the sensitised background by a variant with two DM14-1 domains, Lgd-1-1-C2, was only partial, albeit better than with Lgd-1-C2 (Figure 2H–I’ compared with Figure G,G’). These results indicate that the functional redundancy of the onDM14s is only partial and both domains contribute in a unique manner to the full activity of Lgd. Only the interplay of both domains achieves the full activity of Lgd. Note that the rescue of the sensitised background is better than in the case of Lgd-1-C2 or Lgd-3-4-C2 (comparing I with G and E).

### 3.2. Analysis of the KARRxxR Motif of the onDM14 Domains of Lgd

We previously identified the positively charged surface generated by the KARRxxR motif of the onDM14s as essential for the electrostatic interaction with Shrub [21]. It is also conserved in the onDM14s of Lgd1 and Lgd2 (Figure 3A,B). We wondered whether the AAs of the motif only provide charge, or have an additional structural function. 

To answer this question, we replaced K387 with R and R389, R390, R393 of DM14-3 with K (Figure 3C–I). To remove the functional redundancy between the onDM14s, we used the truncated Lgd-3-4-C2 variant, which can rescue the *lgd* mutant, but not the sensitised genetic background [21,27] (Figure 2D–E’). 

We found that K387 could be replaced by R without any obvious loss of activity, indicating that only a positive charge is required at this position of DM14-3 (Figure 3C,C’). Interestingly, the rescue of the sensitised background by Lgd-3K387R-4-C2 was slightly better than that of Lgd-3-4-C2, indicated by the weaker expansion of the expression domain of Wg and the more restricted expression of Gbe+Su(H) (Figure 3D,D’ compared with Figure 2D–E’). Thus, the introduction of a stronger positive charge at this position is beneficial. In the crystal structure, K387 contributes to the interaction with Shrub via the electrostatic interaction with E90 of Shrub [21] (Figure 3A). AlphaFold2 additionally predicts that K387 makes contact with E382 of DM14-3. The replacement of K387 by R is likely to enhance this dual structural support. The individual replacement of the Rs by Ks leads to only a partial rescue of *lgd* mutants (Figure 3E–G’). This finding indicates that the Rs are essential for function and cannot be replaced by similar charged Ks. 

The A in the KARRxxR motif is conserved in all onDM14 domains of the *Drosophila* and mammalian orthologs of Lgd. We found that an exchange of A388 by G does not affect the ability of Lgd-3-4-C2 to rescue *lgd* mutants, but is unable to rescue the sensitive background (Figure 3H,I). The structure prediction suggests that the side chain of A388 is located in the area between the two helices of the helical hairpin of DM14-3 and it is therefore likely that it stabilises the hairpin via hydrophobic interactions with the AAs of the opposing helix. 

### 3.3. The KARRxxR Motif of the onDM14 Domains Is Not Sufficient for the Function of Lgd 

This work as well as previous ones show that the KARRxxR motif in the onDM14s is an essential feature for the function of Lgd. In order to test whether it is sufficient for the function of the onDM14s, we introduced this motif into DM14-4 of Lgd-4-C2, a variant with only DM14-4. Surface potential analysis showed that the introduction strongly increases the positively charged surface of the even-numbered DM14s (Figure 4A–C’). Nevertheless, Lgd-4-C2-KARRxxR was unable to rescue the *lgd* phenotype (Figure 4F–G’). Moreover, even a variant containing only the two even-numbered DM14 domains with the KARRxxR motif inserted (Lgd-2KARRxxR-4KARRxxR-C2) failed to achieve rescue (Figure 4H–I’). Note that the introduction of the KARRxxR motif into the even-numbered DM14s has no negative effect on the overall structure of Lgd, since a full-length Lgd variant where the KARRxxR motif was inserted into both even-numbered domains, Lgd-1-2KARRxxR-3-4KARRxxR-C2, completely rescued *lgd* mutants (Figure 4J,J’). These results indicate that although essential, the KARRxxR motif is not sufficient for the function of an onDM14 domain.

### 3.4. Analysis of the Meaning of AAs Conserved in All DM14 Domains

The comparison of the DM14 domains of *Drosophila* revealed six other AAs that are highly conserved at the same positions in all four DM14 domains of Lgd (Figure 5A). Out of these six AAs, four are also conserved in the DM14 domains of mammalian LGD1 and LGD2, while the other two are conserved in most domains, or replaced by similar AAs (Appendix A). In the structure predictions of AlphaFold2, these AAs are engaged in interactions within the helical hairpin that are likely to stabilise the hairpin (Figure 5J). We tested their importance for the function of DM14-3 in Lgd-3-4-C2.

A proline (P)-rich sequence follows at the C-terminus of all DM14 domains, with one P absolutely conserved. It appears to interact with a conserved arginine (R368 in DM14-3) and two tyrosines (Y372 and Y398) in the hairpin structure of DM14-3 (Figure 5A). We found that the replacement of P417 by A in DM14-3 of Lgd-3-4-C2 (Lgd-3P417A-4-C2) abolished its rescue ability, indicating that it is essential for the function of DM14-3. The replacement of one of the conserved predicted interaction partners of P417, R368, by A had no effect on the rescue ability of Lgd-3-4-C2, indicating that it is not absolutely required for function, although strongly conserved. However, the ability of Lgd-3R368A-4-C2 to rescue the sensitive background was reduced compared to Lgd-3-4-C2, indicating a reduction in the activity of the variant (Figure 5G, compared with I). Hence, R368 weakly contributes to the full function of DM14-3. In the structural prediction of AlphaFold2, R368 is located in the first helix of DM14-3 and interacts with Y398 of the second helix and G422 at the C-terminal end of the domain (Figure 5J). It is likely that these interactions contribute to the integrity of DM14-3.

Our analysis also showed that the conserved K380, K388 and G408 of DM14-3 are not absolutely required for the function of Lgd-3-4-C2, as the corresponding variants with A replacements could rescue the *lgd* mutant phenotype (Figure 5C–E). However, in the case of K380, the rescue of the sensitive background was again weaker than that of Lgd-3-4-C2, indicating a reduction in activity (Figure 5H, compared with I). K380 appears to interact with E376 within helix1 of DM14-3 (Figure 4J). It is likely that these interactions contribute to the structural stability of the DM14-3 hairpin structure. Altogether, the analysis indicates that besides the strong contribution of P417, R368 and K388 also weakly contribute to the activity of DM14-3, probably by stabilising its hairpin structure.

### 3.5. An Important Function of the C-Terminal Regions for the Stability of Lgd

A recent study reported the atomic structure of the region of the Lgd C-terminal to DM14-4 [14] (Figure 6A). It revealed the existence of an additional helical hairpin similar to the DM14 domains, just in front of the C2 domain, and a ß-sheet structure formed by the region between the region of the extreme C-terminus and the region behind the additional hairpin (Figure 6A,B). These elements form a rigid structure which appears to align the C2 domain. Although similar in structure, the helical hairpin lacks the positive patch of DM14-3 (Figure 6C). To test the importance of these regions, we generated several Lgd variants (Figure 6D). (1) We deleted the region betw.een DM14-4 and the C2 domain (LgdΔ558-664). (2) We replaced this region by a flexible long LAP-tag (Lgd-LAP). (3) We deleted the extreme C-terminus, after the C2 domain (LgdΔC). We found that all three variants rescued *lgd* mutants only very marginally, revealing the importance of the corresponding deleted regions (e.g., see Figure 6E–G’). To further explore the reason for the strong loss of function of the variants, we performed a Western blot analysis to monitor their expression. We found that all variants were not detectable on Western blot, suggesting that they are not properly expressed (Appendix A). Thus, the regions forming the additional hairpin and the ß-sheet are required for the stability of Lgd.

### 3.6. Lgd of C. elegans has Only One Functional Odd-Numbered DM14 Domain

Recently, it has been shown that the Lgd ortholog of *C. elegans* (C.e.Lgd) functions in a similar way to Lgd [7]. Alphafold2 predicts four DM14-like helical hairpins in C.e.Lgd. However, the sequence comparison reveals that only DM14-1 has the KARRxxR motif of a onDM14 domain of Lgd, since the KARRxxR motif DM14-3 of C.e.Lgd is replaced by KMKMNMR (Figure 7A). Our analysis predicts that such changes lead to the inactivation of the DM14-3 and that C.e.Lgd therefore possesses only one functional onDM14. If true, C.e.Lgd would behave like a variant with only one odd-numbered DM14 domain, such as LgdΔDM14-1+2; i.e., it would rescue the *lgd* mutant, but not the sensitised background [21,27]. This is exactly what we found. The presence of one copy of C.e.Lgd, expressed under the control of *lgd*P, rescued the *lgd*-null mutant phenotype completely, giving rise to flies with normal-looking imaginal discs (Figure 7B,B’). However, it fails to rescue the sensitised background (Figure 7C,C’). Thus, C.e.Lgd appears not to possess the functional redundancy observed for other Lgd family members.

### 3.7. Functional Analysis of the N-Terminal Region before DM14-1

All members of the Lgd family have a relatively long N-terminus region prior to DM14-1. In Lgd, this region encompasses 135 AAs. The functional significance of this extreme N-terminus is not clear. Therefore, we generated a variant with this region deleted, LgdΔ1-135 expressed under the endogenous promoter, and performed rescue assays (*lgd*P-lgdΔ1-135-HA). We found that the presence of one copy of LgdΔ1-135 completely rescues *lgd* mutants, as well as the sensitive background, completely (Figure 7D,E). Thus, although present in all Lgd orthologs, the N-terminus before DM14-1 appears not to contribute to the function of Lgd.

## 4. Discussion

Lgd is an important regulator of the ESCRT machinery, which is required for the full activity of the ESCRT-III core component Shrub/CHMP4 [6,9,27]. In its absence, the activity of Shrub is reduced by more than 50%. For its function, the direct interaction with Shrub via its two onDM14s is required [6,9,27]. Previous work suggests that the onDM14s act in a redundant manner. We found here that the redundancy of the onDM14s is not complete, suggesting that both onDM14s have to cooperate to achieve the full functionality of Lgd. The mechanism which underlies this cooperativity is unclear at the moment. Formally, it is possible that the cooperativity lies simply in the increase in the affinity to Shrub. We think that this is not the case, since the rescue of the sensitised background by Lgd-1-3-C2 is complete, while it is only partial for Lgd-1-1-C2 or Lgd-3-3-C2 ([21] and this work). These findings argue against a simple affinity reason for the requirement of the two onDM14s and favours the possibility that the onDM14s have unique functions that are cooperatively required. The nature of the unique functions should be a focus of future work, as the onDM14-mediated interaction with Shrub is also conserved in mammals and is therefore a fundamental mechanism mediated by Lgd family members [2].

It was surprising that C.e.Lgd appears to be unique with the lack of a second functional onDM14. We have previously shown that the presence of both onDM14s is important to be able to tolerate unfavourable genetic constitutions, such as double heterozygosity for *shrub lgd* [9]. Thus, it is required for the robustness of the Lgd/Shrub interaction in the situation of reduced levels of both partners. The outstanding unique feature of the onDM14s is the KARRxxR motif. Interestingly, remnants of the KARRxxR motif can still be recognised in the corresponding C.e.DM14-3, suggesting that the functional robustness provided by the two domains in Lgd orthologs of *Drosophila* and mammals was initially present, but lost during evolution of *C. elegans*. It seems that the robustness is no longer required in *C. elegans*. In favour of this notion is that while the loss of function of Lgd is lethal in *Drosophila* or mice (concomitant loss of both Lgds), it is only conditionally lethal in nematodes, as it depends on the feeding conditions [7,28]. This suggests that the importance of Lgd has diminished during nematode evolution.

We found here that the Rs in the KARRxxR motif cannot be replaced by Ks without a reduction in activity. Nevertheless, the rescue of *lgd* mutants by the corresponding Lgd variants with R to K substitutions is much better than that of correspondent previously analysed variants with R to A substitutions [21]. This indicates that the positive charge is one component of the requirement for function, but not the only one. It also appears that the structure of the Rs is important to make the necessary connection to Shrub. Indeed, in the crystal structure, the Rs of the KARRxxR motif are partly engaged in bivalent interactions with AAs of Shrub [21].

The DM14 domain is a unique feature of the Lgd family [18,19]. The comparison of the DM14s of Lgd identified six absolutely conserved AAs at adequate positions in all four domains. Four of these AAs are also absolutely conserved in the DM14s of LGD1 and LGD2, while the two others are conserved in most cases or replaced by similar AAs. Surprisingly, only the mutation of the absolutely conserved P at the C-terminus of each DM14 abolished the function of DM14-3, while the other AAs lead to no detectable loss of activity, or only to a weak loss of activity, detectable only in the sensitised background. Interestingly, a mutation of the corresponding P in DM14-1 of the mammalian ortholog LGD2 (CC2D1A) has been recently associated with heterotaxy and ciliary disfunction in humans [29]. Our analysis suggests that because of the partial functional redundancy among the onDM14s, the loss of this P results in a weak loss-of-function allele of LGD2, since one of the onDM14s is inactivated. Moreover, they also suggest that the mutation might only cause disease in certain genetic conditions, such as the additional heterozygosity of orthologs of *shrub*, or other mutations that weaken the activity of Shrub. Thus, our results also provide a likely explanation for the rarity of the described mutation [29].

## 5. Conclusions

In conclusion, our work identifies new regions important for the functions of the Lgd/CC2D1A protein family, which are important regulators of the activity of the ESCRT machinery. We identify the KARRxxR motif as crucial for the function of the odd-numbered DM14 domains, which function redundantly in the interaction with the ESCRT machinery. Moreover, we have analysed the importance of amino acids that are conserved among the family-defining DM14 domains. 

## Figures and Tables

**Figure 1 cells-13-01174-f001:**
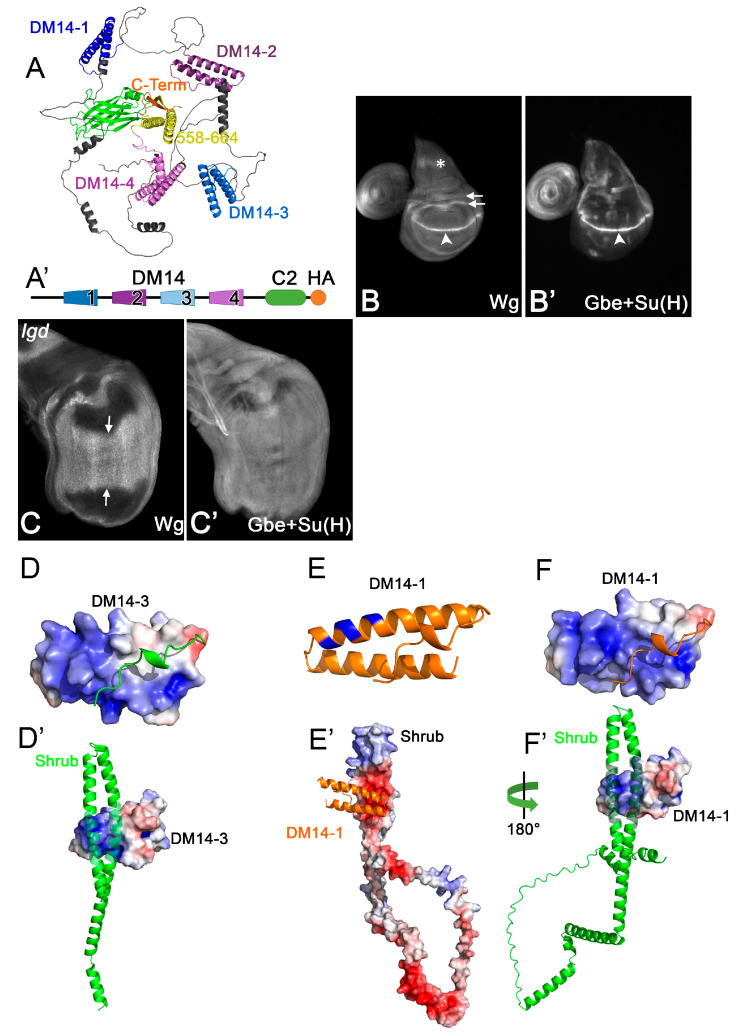
(**A**) Structure of Lgd predicted by AlphaFold2. It consists of four tandem repeats of the DM14 domain followed by a helical hairpin, which lacks the hallmarks of a DM14 domain, and the C2 domain. (**A’**) Schematic representation of Lgd used throughout this work. (**B**–**C’**) Expression of the Notch targets Wg and Gbe+Su(H)-lacZ to measure the activity of the Notch pathway in wildtype (**B**,**B’**) and *lgd* mutant (**C**,**C’**) wing imaginal discs. The arrowhead in (**B**) points to the expression of Wg along the dorsoventral compartment boundary of the wing anlage, which is under control of the Notch pathway and is dramatically expanded in *lgd* mutants (**C**, arrows). The asterisks and arrows in (**B**) highlight expression domains of Wg which are independently of Notch signalling. The expression of Gbe+Su(H) changes from a distinct pattern in wt discs to a uniform staining in *lgd* mutant discs, indicating the uncontrolled activation of the Notch pathway in all disc cells in *lgd* mutants (**C**,**C’**). (**D**) Charge surface distribution of DM14-3, as determined by the crystal structure, reveals an extended positively charged surface (blue). (**E**,**F**) Modelling of DM14-1 reveals a helical hairpin (**E**) with a similar positively charged surface (blue) to DM14-3 (**F**). (**D’**) Binding of DM14-3 of Lgd to a fragment of Shrub encompassing AAs 10-143 out of 226, as determined by [21]. DM14-3 binds via its positive surface to the negative surface of Shrub. The negative surface of Shrub is also required for its homo-polymerisation. (**E’**,**F’**) Modelling of DM14-1 to full-length Shrub predicts a similar interaction as determined for DM14-3. Modelling was carried out using Alphafold2-Multimer Simulation. Surface potential calculations were performed with APBS-electrostatics by [22].

**Figure 2 cells-13-01174-f002:**
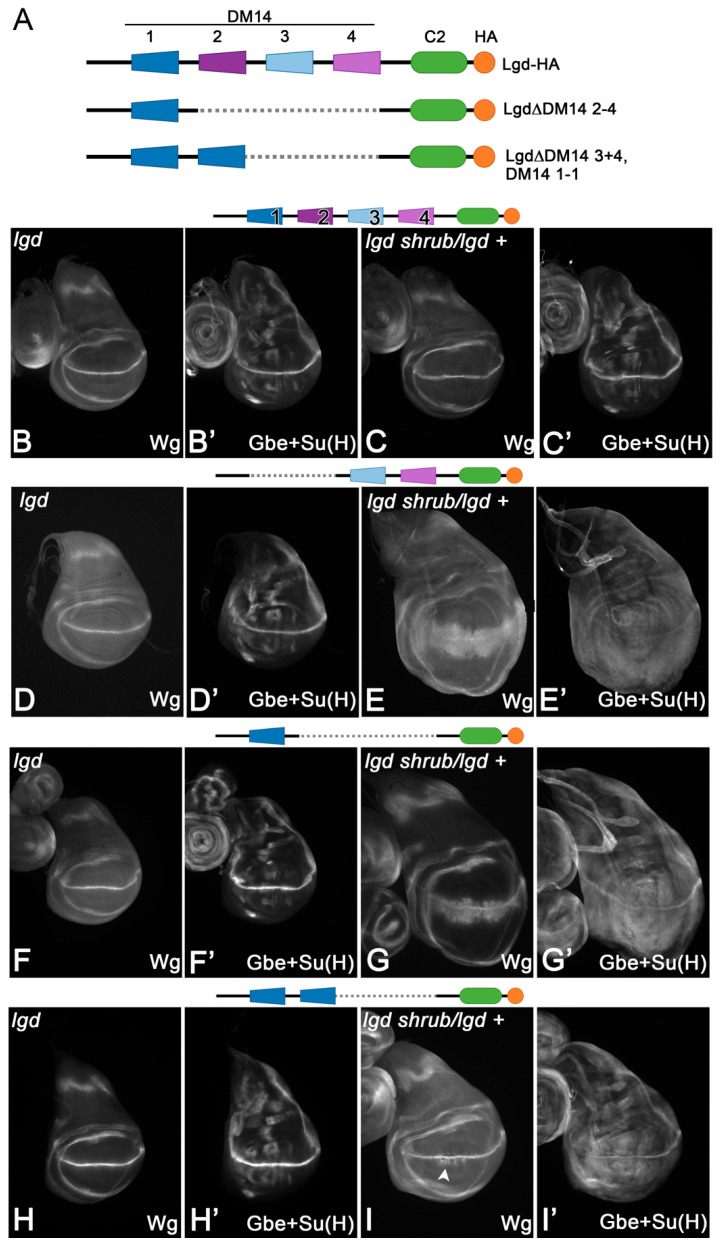
Partial redundancy among the two onDM14 domains of Lgd. (**A**) Schematic representation of the Lgd deletion variants tested. (**B**–**C’**) The presence of one copy of full-length Lgd (Lgd-1-2-3-4-C2) expressed under control of the endogenous promoter (*lgd*P) completely rescues *lgd* mutants (**B**,**B’**) and also the sensitive background (**C**,**C’**). (**D**–**E’**) In contrast, the truncated Lgd-3-4-C2, with only one onDM14, completely rescues the *lgd* mutant (**D**,**D’**), but not the sensitive background, indicating a reduction in activity (**E**,**E’**). (**F**–**G’**) A similar behaviour is observed in the case of Lgd-1-C2. (**H**–**I’**) Even Lgd-1-1-C2, which has two DM14-s domains, is not able to completely rescue the sensitive background indicated by the weak ectopic expression of Wg close to the D/V boundary (arrowhead).

**Figure 3 cells-13-01174-f003:**
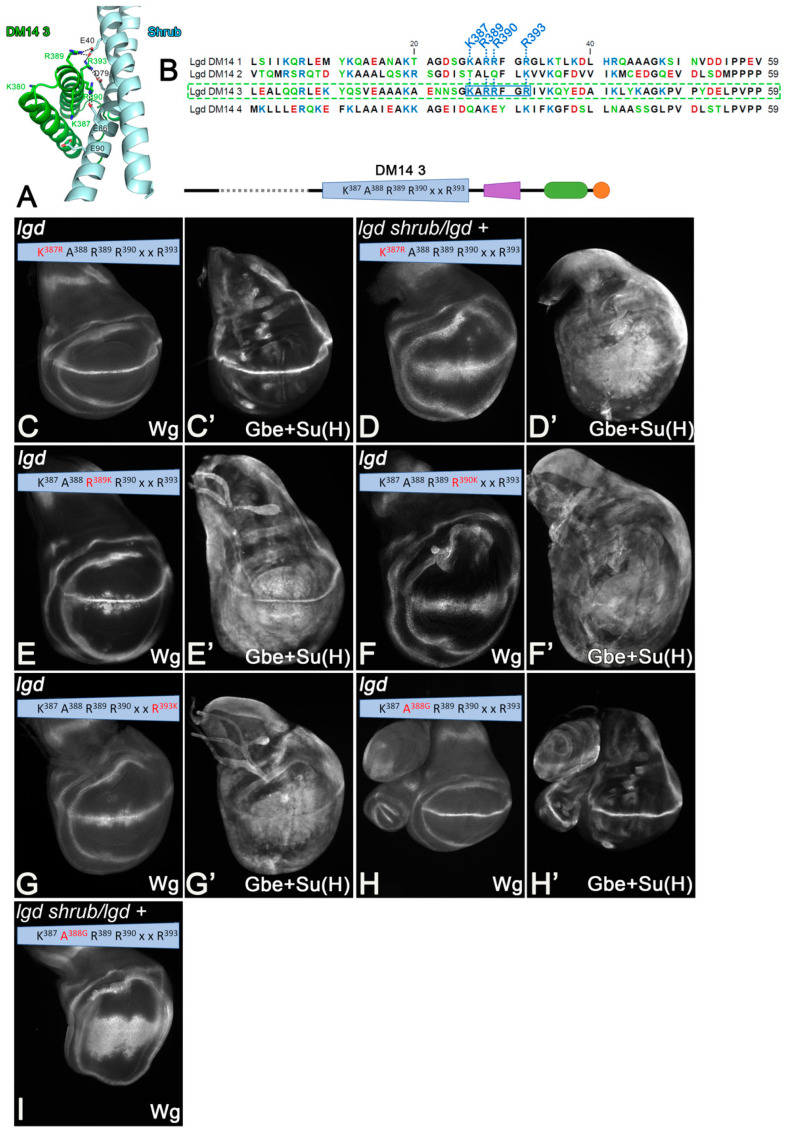
Identification and analysis of the KARRxxR motif of the onDM14s. (**A**) Crystal structure of DM14-3 in complex with Shrub reveals that DM14-3 interacts via a positive surface (from [21]). (**B**) Sequence comparison of the four DM14 domains of Lgd reveals that the KARRxxR motif, which is unique to the onDM14, forms the positive surface required for interaction. This motif is also conserved in the onDM14s of the mammalian Lgds (see Appendix A). (**C**–**I**) The rescue abilities of Lgd-3-4-C2 variants with conservative single AA exchanges of the KARRxxR motif. Compare with Figure 2D–E’.

**Figure 4 cells-13-01174-f004:**
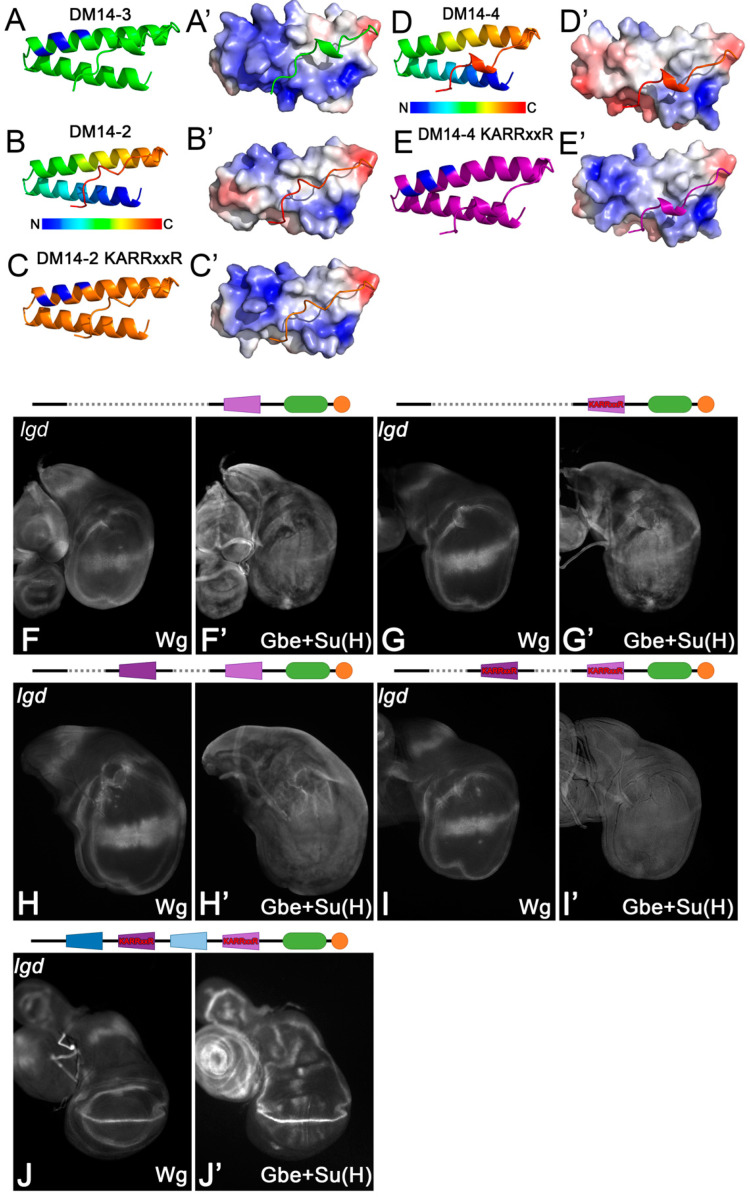
The KARRxxR motif is not sufficient to provide the activity of the onDM14s. (**A**–**E’**) Modelling of the change in the surface potential of the DM14-2 and DM14-4 after introduction of the KARRxxR motif. Calculations were performed with APBS-electrostatics [22]. The surface becomes very similar to DM14-3. (**F**,**F’**,**H**,**H’**) The truncated variants Lgd-4-C2 and Lgd-2-4-C2 only weakly rescue *lgd* mutants (compared with Figure 2B–C’). (**G**,**G’**,**I**,**I’**) The rescue abilities of these two variants are not improved by the introduction of the KARRxxR motif into the DM14s. (**J**,**J’**) The rescue ability of a full-length Lgd is not altered if the KARRxxR motif is additionally introduced to the even-numbered DM14s (compared with Figure 2B–C’).

**Figure 5 cells-13-01174-f005:**
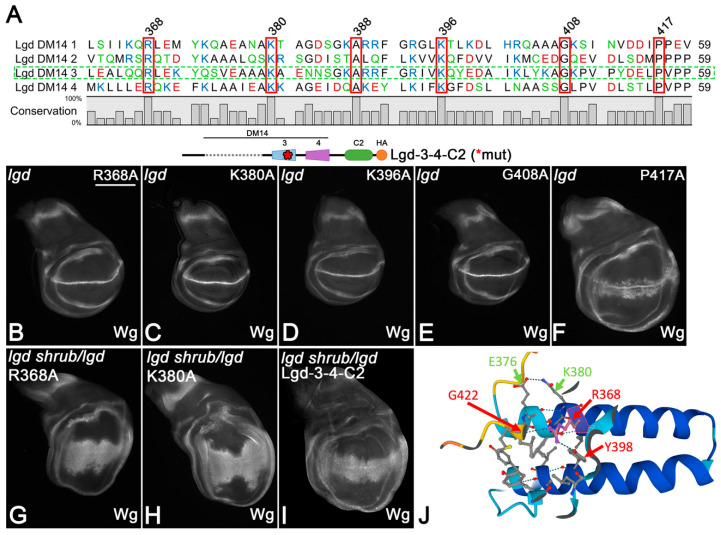
The meaning of the AAs conserved in all DM14 domains. (**A**) Sequence comparison of all DM14 domains of Lgd. The conserved AAs are framed in red. The location of these AAs in DM14-3 are shown in (**J**). (**B**–**F**) Rescue abilities of one copy of Lgd-3-4-C2 with individual exchanges of the conserved AAs in DM14-3. * indicates that DM14-3 is mutated. Only the exchange of P417 with A abolishes the ability of Lgd-3-4-C2 to rescue *lgd* mutants. (**G**–**I**) The rescue of the sensitised background by Lgd-3R368A-4-C2 and Lgd-3K380A-4-C2 is weaker than by Lgd-3-4-C2, indicating that R368 and K380 contribute to the function of DM14-3. (**J**) Prediction of the interactions engaged by R368 and K380 within DM14-3 by AlphaFold2.

**Figure 6 cells-13-01174-f006:**
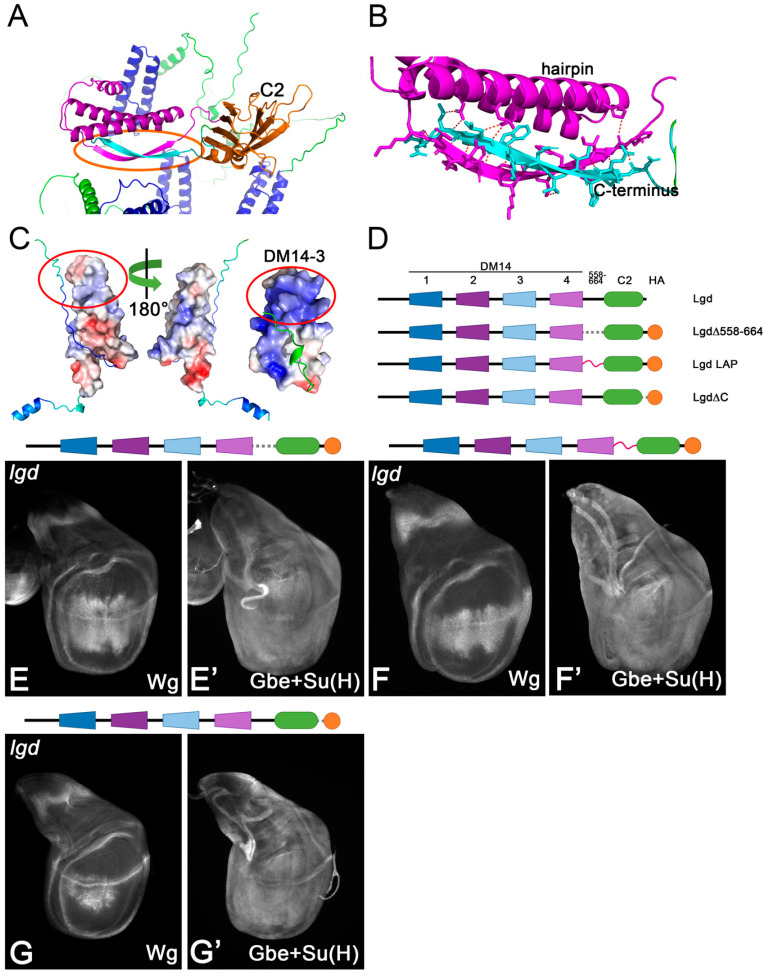
Analysis of the regions of Lgd 558-664 and C-terminal to the C2 domain (see Figure 1A). (**A**) Location of the anti-parallel ß-strand structure forming between the region in 558-664 and the region C-terminal to the C2 domain (framed by the orange ellipse). (**B**) Magnification of the ß-strand region highlighted in (**B**). (**C**) Comparison of the helical hairpin of the region 558-664 with DM14-3. The positive surface charge is in blue, with the negative in red. The helical hairpin does not display a positively charged region comparable to DM14-3 (region encircled in red). (**D**) Schematic representation of the Lgd variants tested to explore the regions described in (**A**–**C**). (**E**–**G’**) Representative wing discs of *lgd* mutants rescued with one copy of the described variants. Structure predictions are performed using Alphafold2.

**Figure 7 cells-13-01174-f007:**
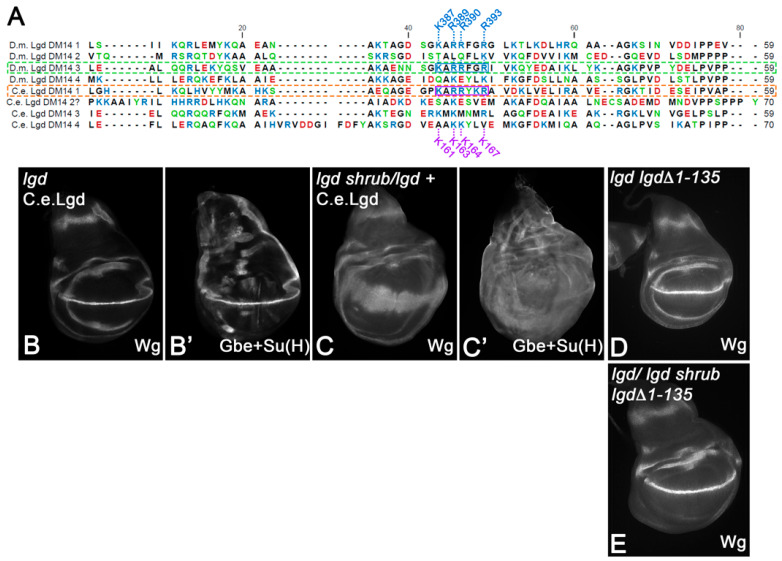
Sequence comparison of the DM14 domains of Lgd with C.e.Lgd. The KARRxxR motif is only present in DM14-1 of C.e.Lgd. (**A**) Comparison of the regions containing the DM14 domains of Lgd with that of *C. elegans*. The question mark for *C. elegans* DM14-2 highlights the strongly deviating sequence of this domain in comparison to the sequence of DM14-2 in other orthologs. (**B**–**C’**) C.e.Lgd can rescue *lgd* mutants (**B**,**B’**), but not the sensitised background (**C**,**C’**). (**D**,**E**) Rescue of *lgd* mutants (**D**) and the sensitised background (**E**) with one copy of *lgd*P-lgdΔ1-135-HA.

## Data Availability

All data are included in the manuscript or Appendix A.

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
