# Peer review of "Structural Analysis of the ESCRT-III Regulator Lethal(2) Giant Discs/Coiled-Coil and C2 Domain-Containing Protein 1 (Lgd/CC2D1)"

_cells, 2024, doi:10.3390/cells13141174_

Round 1

Reviewer 1 Report

Comments and Suggestions for Authors

In their paper, Breuer et al present an in vivo analysis of the functionality of Drosophila Lethal(2) giant disc (Lgd) variants. Lgd acts in flies and worms to support the function endosomal sorting required for transport (ESCRT) -III components for multivescicular endosome (MVE) sorting. By expressing the variants on a Lgd null mutant background or in a Lgd null mutant background that is also heterozygous for Shrub, the fly ESCRT-III component Vps20/Snf, the author are able to also asses which variant rescues and is robust enough to support function when Shrub and Lgd are genetically limiting. The very detailed structure-function analyses in the study return precise knowledge about the role of the DM14 domains of Lgd. 

The paper is solid and well written and uses a meticulous approach to dissect the role of residues of the odd number DM14 of Lgd, that based on a previous analysis by the same authors are the ones sufficient for function. While very restricted in the scope, the study contains interesting and valuable data that provide a solid base to understand the elusive function of Lgd in promoting ESCRT-III activity. I propose a few areas of improvement below:

  1. If I understand correctly, the rescues are based on one Lgd copy driven by its own promoter. While rescues with wt Lgd indicate that one copy is sufficient for function, this could make Lgd limiting in the sensitized background (lgd null, one rescue copy, Shrub heterozygous). I was wondering whether a full complement (2 copies) of Lgd rescue in Shrub heterozygous would result in full rescue for some of the variants. Considering that Lgd and Shrub are on chr II and the rescue constructs are on chr. III this could be done at least for some variants.
  2. While the interpretation of the results relative to the extent of rescue are reasonable, it would useful to devise a method to quantify the effects in discs. One possibility would be to determine the distance from the DV boundary for Wg. For Gbe+Su(H), which is more broadly distributed when Ldg is non functional, I suggest the mean grey intensity, normalized over the pouch area. Also, disc size, which is a read-out of Notch overactivation could be quantified. This would allow to illustrate the data taking into account the average values over a pool of discs, rather than presenting representative examples only. 
  3. Together with Notch signaling effect, it is conceivable that the presented discs could show defects in endosomal degradation of Notch. Considering that inexpensive monoclonals against Notch that work very well exist, would it be useful to also determine the extent of endosomal puncta in the genetic background analyzed. If this correlates well with the reporters, it could be used in the first figures only to illustrate the link between endosomal defect and signaling alterations. Perhaps, together with quantitative analysis of reporters and tissue proliferation, this analysis could provide a better way to rank levels of rescue?
  4. Given the complex genotypes, a table of genotypes shown in figures as part of supplementary methods would be useful to orient the readers
  5. I am not sure it is fair to state in the methods that primers for Gibson cloning are available upon request. They should be presented in a supplementary table. 
  6. There’s a few typos here and there (line 181: Fig 3A,B?; line 225: additional a to delete; line 283: Fig5S1?; line 427: with?) and some of the English could be improved. 

Comments on the Quality of English Language

see point 6 above

Reviewer 2 Report

Comments and Suggestions for Authors

I have completed a thorough review of the manuscript titled "Structural analysis of the ESCRT-III regulator Lethal(2) giant discs/ Coiled-coil and C2 domain-containing protein 1 (Lgd/CC2D1)." The authors perform a comprehensive analysis of the domains and amino acid functions in the Lgd protein from Drosophila. Overall, the manuscript demonstrates a satisfactory level of English language proficiency. However, to enhance its quality and readiness for publication, I recommend the following minor revisions.

Supplementary Names: Please rename the supplementary figures appropriately. Specifically, correct "Fig. 5S1" and "Fig. 6S1."

Typographical Errors: Remove the question mark in line 239 (currently written as "Fig. 4?").

Figure Labeling: Be careful with the figure labels. In line 249, what is currently referred to as "Figure 4J" should be "Figure 5J."

Hypothesis Presentation: Lines 280-283 present only a hypothesis since it is not definitively proven. Please change this to a conditional form.

Figure 7 Clarification: In Figure 7, please clarify the meaning of the question mark in "C.e.Lgd DM14 2."

Protein Structure Source: Indicate whether the protein structure was predicted by AlphaFold or obtained from the AlphaFold database.

References Consistency: Ensure that all references are homogenized. Some include PMID numbers, others URLs, and some have no identifiers. Specifically, review reference 28, ensuring it includes the PMID 7729581.

Domain Sequence: In line 437, the domain should be presented as "KARRxxR."

Figure Representation: In line 283, consider using Figures 6A-B instead of Figure 5S1, as they may be more representative.

Comments on the Quality of English Language

the manuscript demonstrates a satisfactory level of English language proficiency. 
